# The quantity and composition of household food waste: Implications for policy

**Drajat Martianto**[1☯], **Rian Diana**[2,3☯]*, **Yayuk Farida Baliwati**[1☯], **Dadang Sukandar**[1☯], **Agung Hendriadi**[4☯]

1 Department of Community Nutrition, Faculty of Human Ecology, IPB University, Bogor, Indonesia,
2 Department of Nutrition, Faculty of Public Health, Universitas Airlangga, Surabaya, Indonesia, 3 Program of Nutrition Science, Graduate School, IPB University, Bogor, Indonesia, 4 Agroindustry Research Center, National Research and Innovation Agency, Jakarta, Indonesia

☯ These authors contributed equally to this work.
* rian.diana@fkm.unair.ac.id

**Data Availability Statement:** The research data available are at https://data.brin.go.id/dataset.xhtml?persistentId=hdl:20.500.12690/RIN/50068E.

**Funding:** The authors thank the Neys-van Hoogstraten Foundation and IPB University [Rian

## Abstract

Studies on food waste in Southeast Asia are currently limited, with a notable absence of comparative analyses investigating the volume and composition of food waste in urban and rural areas through direct measurement. This study aimed to analyze the differences in the quantity, composition, and drivers of household food waste between urban and rural areas. Household food waste was assessed through waste compositional analysis for food and diaries for beverages. This cross-sectional study included 215 households in Bogor Regency, Indonesia. Comparisons between the two areas were performed using an independent t-test. The average of household food waste in Bogor Regency was 77 kg/cap/year (edible 37.7%, inedible 62.3%). Household food waste was higher in urban areas (79.4 kg/cap/year) than in rural areas (45.8 kg/cap/year) (p<0.001). Cereals, tubers and their derivatives (especially rice) and vegetables were the major contributors to edible food waste, whereas fruits were the main contributors to inedible food waste in both areas. Food waste drivers were spoilage/staleness/moldiness, changes in texture, short shelf life, cooking too much, and plate leftovers. Households in urban areas had a higher quantity of food waste and disposed of more edible food than those in rural areas. Meanwhile, the drivers of food waste generation were similar in both areas. Understanding the quantity, composition, and drivers of household food waste is pivotal for developing effective awareness campaigns and fostering behavioral changes to prevent household food waste.

## Introduction

Food loss and waste refer to a reduction in both the quantity and quality of food intended for human consumption along the food supply chain [1]. Food waste (FW) occurs in retail (e.g., supermarkets and traditional markets), food services (e.g., restaurants, catering, canteens, and home consumption), households, and individuals [2–6].

The United Nations Environment Programme (UNEP) estimated that 17% of the food produced globally was discarded or not consumed. In 2019, the global average FW reached 121

Diana, grant numbers: 01 /NHF-IPB l 2022] for funding this study. The funders did not involve in the study design, data collection, data analysis and preparation of the manuscript or decision to publish.

**Competing interests:** The authors declare no conflict of interest for this publication.

kg/capita/year, equivalent to 931 million tons of food discarded at retail and consumer levels. Furthermore, households accounted for 61% of the total FW, while food services and retail contributed 26% and 13%, respectively [7].

Given Indonesia's status as the fourth-most populous country in the world, understanding its food loss and waste trends become crucial. A study on food waste study in Indonesia over the two decades (2000–2019) revealed a concerning increase from 39% to 55%, equivalent to 5–19 million tons per year. Notably, 80% of this waste originated from households, with 44% being edible food [8]. Despite this significant issue, there remains a dearth of detailed data on the characteristics of FW in Indonesia, particularly through direct measurement. Halving global FW per capita at the retail and consumer levels is one of the targets of the Sustainable Developments Goals 12. To achieve this target, accurate FW measurements are essential for identifying the contributing factors.

Studies on household FW have been conducted in many countries, especially developed countries in Europe [9–14], America [15, 16], Canada [17, 18], and China [19]. The methods used to quantify food waste vary, including direct measurements such as waste composition analysis [10–13, 18] and waste audits [17], as well as indirect measurements like modeling [14, 16] and questionnaires [9, 19]. These studies also examined nutrient loss [15], factors related to food waste such as family characteristics [13, 19], dietary quality [16, 18], and consumer behaviors [9]. However, a disproportionate number of studies have been conducted in urban areas than rural areas [7]. Additionally, research on household FW quantity and composition in middle-income countries, particularly Southeast Asia, is limited. While, some studies utilize direct measurements of household FW, there is a lack of data on FW categorized by food types and groups, or comparisons between urban and rural areas [20–23].

A systematic review revealed that no study in Southeast Asia has compared the quantity and composition of FW in urban and rural areas using direct measurement [24]. Therefore, this study aims to address this gap by investigating the following research question: How much food waste is generated, what are the compositions and drivers of household food waste in urban and rural areas, and what are the differences in household food waste generation and composition between these two areas in Indonesia?. The current study analyze the disparities in the quantity, composition, and drivers of household FW between urban and rural areas in Indonesia.

## Methods

Household food waste measurements were conducted from 4–30 October 2022 using two methods: the waste compositional analysis (WCA) for solid waste (food) and diaries for beverages. The WCA method was performed for eight consecutive days in accordance with the Indonesian National Standard (SNI) 19-3964-1994 for the collection and measurement of samples of urban waste generation and composition. Food waste from home trash bins was collected daily, sorted, and weighed per food type using a QME brand digital scale (QM-5140), with a precision of up to 2 g. This method is suitable for mixed solid waste, such as households [25, 26], and its accuracy is high because of direct weighing and waste sorting by type and composition, eliminating the possibility of under-reporting [25].

The diary method was used to quantify drink waste, which was carried out over a seven-day period. Household members were requested to record the type and quantity of discarded beverages as well as the reason for discarding them. However, this method has several limitations, such as participant forgetting to record, unrecorded beverages, inaccurate estimation of discarded drink quantities, and changes in behavior during the recording period [27, 28]. To mitigate these limitations, clear instructions and a well-designed recording system were provided [28]. Drink waste is typically disposed of down the sink, rendering the WCA method

unsuitable for its measurement. In this study, participants were instructed to record discarded beverages daily, and were provided with a 100 ml measuring cup to quantify the volume of the beverage waste. Additionally, a structured questionnaire was administered to housewives to investigate household characteristics and the drivers of household food waste generation.

The sample size was determined in accordance with SNI 19-3964-1994, which considered the population size, number of households, and average number of household members in the two selected sub-districts. A total of 215 households were selected for the FW measurements: 150 in urban areas and 65 in rural areas. Multistage random sampling was employed to select the households. In the first stage, sub-districts with urban and rural characteristics were selected. In the second stage, villages or urban wards with the largest population were selected, and then neighborhoods within each village were randomly chosen. Finally, household samples were randomly selected from the list of households obtained from these selected neighborhoods.

This study was conducted in two areas of Bogor Regency, Indonesia, between October and September 2022. The Cibinong Sub-district was selected as the urban area, as it is the largest population center in the regency, and 100% of its area has urban characteristics. The Sukajaya sub-district was chosen as a rural area because 63.6% of its area has rural characteristics.

The materials and tools used to measure FW using the WCA method were plastic bags, digital scales 0–400 kg, gloves, and recording forms. Diary form-filling guidelines, pens/pencils, and measuring cups were used to measure the diary method.

FW was calculated by weighting the number of households in each neighborhood and the number of family members in each household. The differences in quantity and composition of household food waste between rural and urban areas were analyzed using an independent samples t-test, which was conducted using IBM SPSS 21.

To calculate the average FW in Bogor Regency in kg per capita per year, FW data were extrapolated from the FW quantity in urban and rural households. This method was adapted from the National Zero Waste Council, Canada [29]. In this study, the adaptation involved calculating the quantity of food waste in kilograms per capita per household in urban and rural areas, with extrapolation adjusting for the urban and rural populations proportionally. Therefore, the calculation for the average FW at the regional level was conducted by multiplying the kilograms of FW per household in urban areas by the urban population, adding it to the kilograms of FW per household in rural areas multiplied by the rural population, and then dividing by the total population. The total population of Bogor Regency at the end of 2022 was obtained from the Bogor Regency Population and Civil Registry Office. The total population in Bogor Regency was 5,427,068 people, consisting of 390,047 people in rural areas and 5,037,021 people in urban areas.

This study was approved by the Human Research Ethics Committee of Bogor Agricultural University (number 747/IT3.KEPMSM-IPB/SK/2022). Participants were fully informed of the study's objectives and benefits, and were provided with the right to withdraw at any time without consequences. Written informed consent was obtained from all participants prior to the interview. All data obtained from this research will be treated confidentially, only the research team responsible for this study will be able to access and use it.

## Results

### Household characteristics

The mean ages and educational levels of husbands and wives in urban areas were slightly higher than those in rural areas. Household income and expenditures in urban areas were almost double those in rural areas. (Table 1).

**Table 1. Household characteristics (mean ± SD).**

| Characteristics | Urban | Rural |
|---|---|---|
| Husband's age (years) | 45.7 ± 11.1 | 43.5 ± 10.9 |
| Wife's age (years) | 42.5 ± 10.8 | 39.4 ± 10.1 |
| Husband's education (years) | 10.8 ± 3.3 | 6.3 ± 2.0 |
| Wife's education (years) | 9.8 ± 3.5 | 5.8 ± 1.9 |
| Number of household members (people) | 4.0 ± 1.1 | 4.3 ± 1.3 |
| Income (IDR/cap/month) | 1,237,135 ± 774,449 | 486,903 ± 458,301 |
| Household expenditure (IDR/cap/month) | | |
| Food | 509,096 ± 285,619 | 285,613 ± 119,398 |
| Non-food | 687,102 ± 443,620 | 392,396 ± 239,733 |
| Total | 1,196,198 ± 613,280 | 678,009 ± 296,817 |

## Quantity and composition of FW

This study revealed that the average FW in urban areas (79.4/cap/year) was almost double that in rural areas (45.8 kg/cap/year) (Fig 1). Additionally, the proportion of edible FW was higher in urban areas (38.2%) than in rural areas (23.4%). The findings indicated that urban areas generated more FW per capita than rural areas for edible, inedible, and total FW (p<0.001).

Urban areas generated a greater quantity of FW, both in terms of type and composition, than rural areas (S1 Table). An exception to this trend was observed in inedible FW from fruits and their derivative products as well as inedible FW from cereals, tubers, and their derivative products. Urban households produced a higher proportion of edible FW than their rural counterparts. In urban areas, the most commonly discarded foods were cereals, tubers, and their derivatives (17.9 kg/cap/year), whereas rural areas had a higher disposal of vegetables (4.6 kg/cap/year). Fruit and its derivative products were the most commonly discarded inedible FW in both urban and rural areas, at a rate of 29.9 kg/cap/year and 21.7 kg/cap/year, respectively. Additionally, over 70% of the edible FW in both regions consists primarily of cereals, tubers,

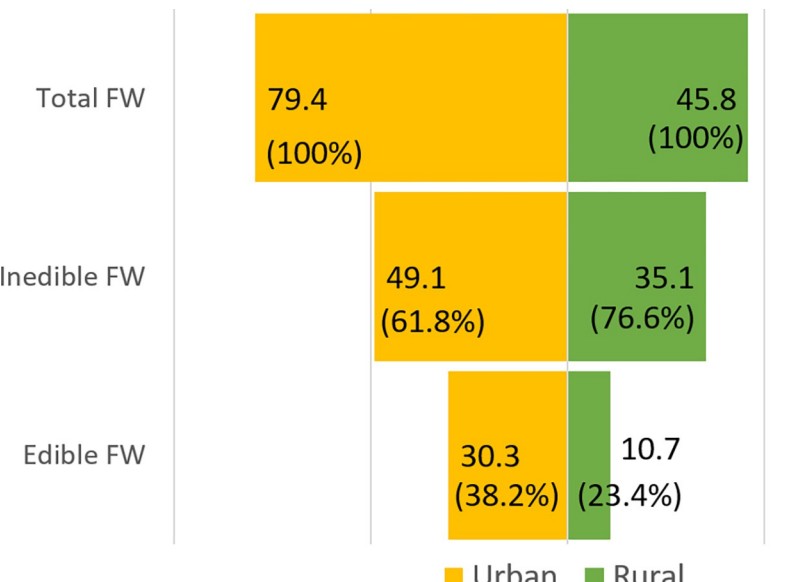

**Fig 1. Mean edible, inedible and total FW in urban and rural areas (kg/cap/year).**

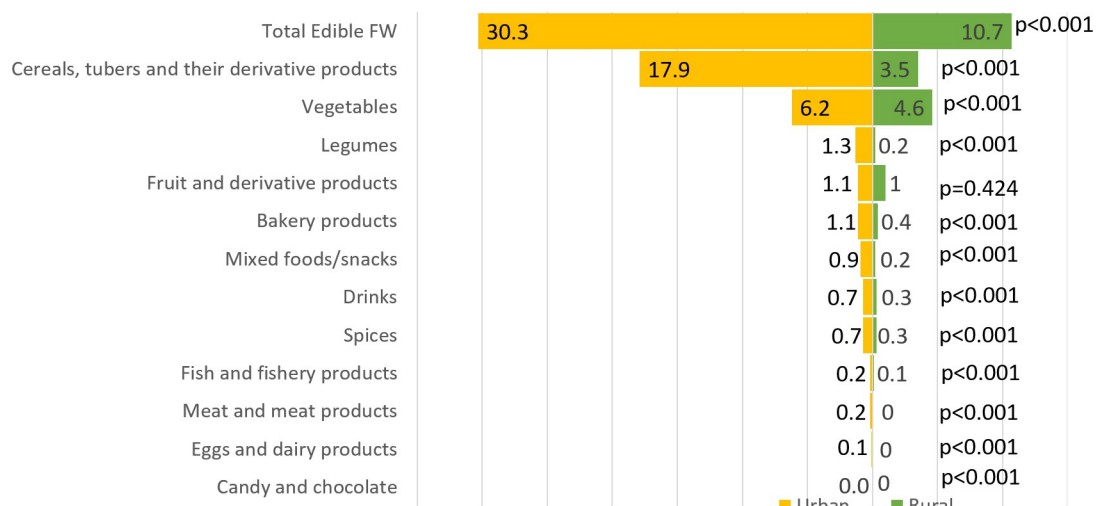

**Fig 2. Mean edible FW by food groups in urban and rural areas (kg/cap/year).**

their derivatives, and vegetables. Conversely, over 70% of inedible FW consisted of fruits, their derivatives, and vegetables (Figs 2 and 3).

The most discarded edible waste per capita annually was rice (14.2 kg), carrots (0.9 kg), Chinese cabbage (0.7 kg), cabbage (0.6 kg), tempeh (0.6 kg), tofu (0.4 kg), guava (0.2 kg), and water apples (0.2 kg). Among the beverages, coffee (0.3 kg) and tea (0.2 kg) had the highest annual per capita waste, with animal food per capita being less than 0.1 kg (S2 Table).

Table 2 reveals that the average household FW in Bogor District is 77 kg/cap/year, consisting of 37.7% edible FW (29 kg/cap/year) and 62.3% inedible FW (48 kg/cap/year). Edible FW was mainly composed of cereals, tubers, and their derivatives at 16.9 kg/cap/year (mainly rice), and vegetables at 9.6 kg/cap/year. Meanwhile, inedible FW was dominated by fruits and their derivative products, accounting for 21.7 kg/cap/year (mainly bananas and mangoes).

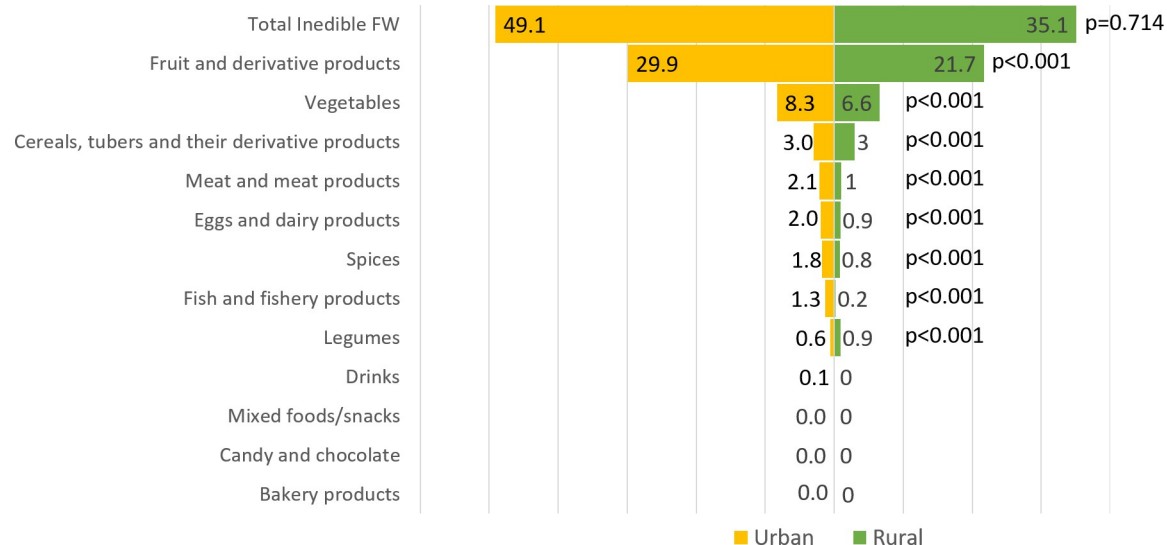

**Fig 3. Mean inedible FW by food groups in urban and rural areas (kg/cap/year).**

**Table 2. Mean and proportion of FW quantity by type and food group in Bogor District (kg/cap/year).**

| Food Groups | *Edible FW* | % | *Inedible FW* | % | Total FW | % |
|---|---|---|---|---|---|---|
| Cereals, tubers and their derivative products | 16.9 | 58.3 | 3.0 | 6.3 | 19.9 | 25.8 |
| Bakery products | 1.1 | 3.8 | 0 | 0.0 | 1.1 | 1.4 |
| Meat and meat products | 0.2 | 0.7 | 2.0 | 4.2 | 2.2 | 2.9 |
| Fish and fishery products | 0.2 | 0.7 | 1.2 | 2.5 | 1.4 | 1.8 |
| Eggs and dairy products | 0.1 | 0.3 | 1.9 | 4.0 | 2.0 | 2.6 |
| Spices | 0.7 | 2.4 | 1.7 | 3.5 | 2.4 | 3.1 |
| Legumes | 1.2 | 4.1 | 0.6 | 1.3 | 1.8 | 2.3 |
| Vegetables | 6.1 | 21.0 | 8.1 | 16.9 | 14.2 | 18.4 |
| Fruit and derivative products | 1.1 | 3.8 | 29.4 | 61.3 | 30.5 | 39.6 |
| Candy and chocolate | 0 | 0.0 | 0 | 0.0 | 0 | 0.0 |
| Mixed foods/snacks | 0.8 | 2.8 | 0 | 0.0 | 0.8 | 1.0 |
| Other foods | 0 | 0.0 | 0.1 | 0.2 | 0.1 | 0.1 |
| Drinks | 0.6 | 2.1 | - | - | 0.6 | 0.8 |
| Total | 29 | 100.0 | 48 | 100.0 | 77 | 100.0 |

## Reasons for throwing away food

Five primary reasons were identified for discarding food: spoilage/staleness/moldiness, changes in texture, short shelf life, cooking too much, and plate leftovers. Sensory characteristics, such as staleness or moldiness and changes in texture, were the main reasons for throwing away rice and vegetables. The fruits were discarded because of moldiness, bruises, and changes in texture. Both animal- and plant-based foods have been discarded for various reasons. Dairy products were often discarded because they had been forgotten in the refrigerator and remained unconsumed. Frequent reasons for discarding meat, fish, and eggs were mold or mildew, and textural changes. Meanwhile, tofu and tempeh were discarded due to cooking too much, changes in texture, and unconsumed. Bread was discarded when it was moldy or exceeded its best-before date. Inappropriate storage temperature and sensory changes (staleness or moldiness, and texture changes) were the main reasons for discarding packaged food (see Table 3). The reasons reported by the respondents in both urban and rural areas were almost identical for all types of food.

## Discussion

The average FW found in this study (77 kg/cap/year) was comparable to the average FW in upper-middle-income (76 kg/cap/year) and high-income (79 kg/cap/year) countries [7]. In

**Table 3. Number of respondents by FW drivers.**

| Food | Spoil/ Stale/ Moldy | Cooking too much | Short shelf life | Changes texture | Plate leftovers | Forgotten | Others |
|---|---|---|---|---|---|---|---|
| Rice | 52 | 53 | 24 | 56 | 57 | 3 | 21 |
| Vegetables | 63 | 48 | 60 | 9 | 27 | 8 | 37 |
| Fruits | 32 | 1 | 3 | 15 | | 5 | 9 |
| Dairy products | 4 | | | 4 | 5 | 10 | 9 |
| Meat, fish, and eggs | 14 | 6 | 6 | 12 | 6 | 1 | 15 |
| Tempeh and tofu | 6 | 25 | 4 | 9 | 9 | 7 | 12 |
| Breas | 31 | 1 | 1 | 1 | | 2 | 9 |
| Packaged foods | 5 | | | 5 | 1 | 3 | 19 |
| Cooked foods | 11 | 2 | 11 | 4 | 1 | | 15 |
| Total | 218 | 136 | 113 | 111 | 106 | 39 | 146 |

contrast, edible FW in upper-middle-income and high-income countries is higher, ranging from 50–65% [10–13, 30] compared with the 37.7% found in this study, which is a lower-middle-income country. The discarded food groups across countries displayed a similar composition, with staple foods, fruits, and vegetables being the main contributors to the total FW. This study aligns with previous research conducted in high-income countries, where staple diets consisting of rice [19] or bread [11, 31–33].

This study revealed that the average FW in urban areas was almost twice as high as that in rural areas (Fig 1), consistent with previous research conducted in lower, middle, and high-income countries [34, 35]. The quantity of FW in urban areas (79.4 kg/cap/year; 38.2% edible) exceeded that in rural areas (45.8 kg/cap/year; 23.4% edible). Thus, household food waste was significantly higher in urban areas than in rural areas (p<0.001). Nevertheless, the composition and reasons for discarding food were similar between the two regions (Table 3).

The most commonly discarded edible FW in both urban and rural areas consist of cereals, tubers and their derivatives, vegetables, and legumes (Fig 2). The main contributors to inedible food waste were fruits and their derivatives, along with vegetables (Fig 3). The results of this study imply that strategies for preventing and reducing household FW can focus on four food groups. In urban areas, education can be concentrated on preventing and reducing edible FW (rice, vegetables, tempeh, and tofu) as well as practical training to reuse or recycle inedible FW (fruits and vegetables). Meanwhile, in rural areas, education can be focus reducing edible FW (vegetables) by improving storing practices and practical training to reuse or recycle inedible FW (fruits).

Various reasons underlie food waste, depending on the type of food (Table 3). Information on food storage, including techniques to extend the shelf life of food and proper food storage practices, is applicable to all types of food. Encouraging behavioral changes in cooking and eating habits is crucial to preventing and reducing food waste, especially for foods such as rice, vegetables, plant-based foods (tofu and tempeh), and cooked foods. Utilizing data-related information on food packaging can help prevent wastage of bread and packaged foods.

The proportion of inedible FW was higher than that of edible FW in both the urban and rural areas. Urban households were more likely to discard edible FW than rural households (38.2% and 23.4%, respectively) (Fig 1). This implies that urban households throw away more food suitable for consumption. Consistent with studies in the European Union and Lebanon, this study found that the quantity of household FW in urban areas was higher than that in rural areas [34, 35].

Household food waste generated in urban areas exceeds that of rural areas, attributed to disparities in food systems, food access, consumer behavior, and food waste utilization or management. In rural settings, self-produced food sources prevail, contrasting with the urban areas where processed food purchases predominate, leading to amplified food waste generation [36]. The growth of modern retail, restaurants, and fast-food outlets in urban areas affects consumer preferences, purchasing behavior, and food accessibility, thereby driving higher food waste rates [37]. Urban residents tend to prefer quick and convenient food preparation, resulting in over-purchasing, high consumption of processed foods, and foods from outside the home (restaurants, fast food outlets, and street food) [36]. This trend contributes to the generation of food waste [36, 38]. On the other hand, many rural households convert their food waste into compost or use it as feed for pets or livestock [39, 40], thereby reducing the amount of FW disposed of in bins [39].

This study found a higher household income and education level in urban areas than in rural areas (Table 1). Income is a major driver to household FW generation [19]. Urban residents have higher incomes but allocate a lower proportion of food expenditure than those living in rural areas, leading to a higher dietary diversity [37]. Moreover, the higher dietary

quality correlates with greater FW, as individuals with a high dietary quality consume larger quantities of vegetables and fruits, which are the most discarded food waste [41].

## Policy implication

Indonesia has established policies and regulations related to food waste; however, there is a notable absence of a roadmap or guidelines for reducing food waste along the food supply chain, particularly at the household level. Consequently, there is a pressing need to develop policies aimed at reducing household food waste, which should include awareness campaigns and behavior change initiatives.

Policies targeting the prevention of household food waste should encompass both preventive measures and food waste management. Prevention involves actions taken before edible food is discarded, whereas management focuses on dealing with food that is no longer suitable for consumption or has already been discarded. Prevention strategies can be implemented through awareness campaigns and initiatives aimed at encouraging behavioral changes. Conversely, managing food waste can entail practices such as reuse and recycling by converting it into feed, compost, and eco-enzymes.

Prevention and reduction strategies should be concentrated in urban areas and focus on food groups that significantly contribute to edible FW such as cereals, tubers, and their derivatives, particularly rice, vegetables, and legumes. Prioritizing the reuse and recycling of inedible food waste, particularly from fruits and vegetables, is essential to prevent its disposal in landfills.

Behavioral changes that emphasize on prolonging the shelf life of food and practicing proper storage can be implemented for all types of foods. Adjusting cooking portions and eating habits can be particularly beneficial for rice, vegetables, plant-based proteins such as tofu and tempeh, and cooked foods. Understanding food packaging data can help minimize food waste, particularly for bread and packaged goods. Finally, reusing food waste for feed and composting inedible fruits and vegetables at home rather than sending them to landfills is an important step in reducing food waste.

## Research limitation

This study was conducted using direct FW measurements (Waste Composition Analysis) over eight consecutive days, ensuring high accuracy through sorting and weighing, based on type and composition. However, it captured only a single point in time and did not include seasonal data. Additionally, the study only measured FW disposed of in household bins and did not consider FW disposed of or given to pets, plants, or other disposal channels. Despite these limitations, this study found that over 82.8% of household food waste was disposed of in the bin, with only 16.7% discarding FW onto plants for composting.

Future studies could be conducted to quantify the household food waste disposed of through various channels (trash bins, sink, animal feed, and compost) at different points in time (seasonal or beginning and end of the month). In addition, research on waste prevention and reduction policies tailored to the characteristics of urban and rural areas, as well as food waste behaviors is needed.

## Conclusion

The quantity of household food waste differs significantly between urban and rural areas, with urban households discarding more edible food than their rural counterparts. Cereals, tubers and their derivatives (especially rice), vegetables, and legumes were the major contributors to edible food waste, whereas fruits were the main contributors to inedible food waste in both

areas. Additionally, the drivers of food waste were found to be similar in both urban and rural areas.

Strategies for food waste prevention and reduction should target urban areas and focus on food groups that contribute significantly to edible food waste, such as cereals, tubers, and their derivatives, particularly rice, vegetables, and legumes. Meanwhile, managing inedible food waste should prioritize the reuse and recycling of fruits and vegetables to prevent their disposal in landfills, especially in rural areas.

Disparities exist in food waste drivers, primarily depending on the type of food rather than the location of the resident. Therefore, awareness campaigns and behavior change initiatives should take into account the main drivers of food waste based on the type of food. Understanding the quantity, composition, and drivers of household food waste is crucial for developing effective awareness campaigns and fostering behavioral changes to prevent household food waste.

## Supporting information

**S1 Table. Mean and standard deviation of FW quantity by type and food group in urban and rural areas (kg/cap/year).**
(DOCX)

**S2 Table. Quantity of frequent edible food waste (mean g/cap/year).**
(DOCX)

## Author Contributions

**Conceptualization:** Drajat Martianto, Rian Diana, Yayuk Farida Baliwati, Agung Hendriadi.

**Methodology:** Dadang Sukandar.

**Supervision:** Drajat Martianto.

**Writing – original draft:** Drajat Martianto, Rian Diana.

**Writing – review & editing:** Drajat Martianto, Rian Diana, Yayuk Farida Baliwati, Dadang Sukandar, Agung Hendriadi.

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
