## [Decision Letter · Decision Letter 0]

18 Mar 2024

PONE-D-24-03754The quantity and composition of household food waste: implications for policyPLOS ONE

Dear Dr. Diana,

Thank you for submitting your manuscript to PLOS ONE. After careful consideration, we feel that it has merit but does not fully meet PLOS ONE’s publication criteria as it currently stands. Therefore, we invite you to submit a revised version of the manuscript that addresses the points raised during the review process.

We look forward to receiving your revised manuscript.

Kind regards,

Fabien MUHIRWA

Academic Editor

PLOS ONE

“The authors thank the Neys-van Hoogstraten Foundation and IPB University [Rian Diana, grant numbers: 01 /NHF-IPB I 2022] for funding this study. The funders did not involve in the study design, data collection, data analysis and preparation of the manuscript or decision to publish.”

“The authors thank the Neys-van Hoogstraten Foundation and IPB University [Rian Diana, grant numbers: 01 /NHF-IPB I 2022] for funding this study. The funders did not involve in the study design, data collection, data analysis and preparation of the manuscript or decision to publish.”

Additional Editor Comments:

Dear Authors, Congratulations on the completion of this insightful manuscript, please try to revise the discussion section a bit. And pay close attention to the reviewers' comments.

Reviewers' comments:

Reviewer's Responses to Questions

**Comments to the Author**

1. Is the manuscript technically sound, and do the data support the conclusions?

Reviewer #1: Yes

Reviewer #2: Partly

Reviewer #3: Yes

2. Has the statistical analysis been performed appropriately and rigorously? 

Reviewer #1: Yes

Reviewer #2: Yes

Reviewer #3: Yes

3. Have the authors made all data underlying the findings in their manuscript fully available?

Reviewer #1: No

Reviewer #2: Yes

Reviewer #3: Yes

4. Is the manuscript presented in an intelligible fashion and written in standard English?

Reviewer #1: Yes

Reviewer #2: Yes

Reviewer #3: Yes

5. Review Comments to the Author

Reviewer #1: This study focuses on analyzing the differences in the amount, types, and causes of household food waste between urban and rural areas. It was conducted in Bogor Regency, Indonesia, and involved 215 households. The researchers used waste compositional analysis for food and diaries for beverages to assess household food waste. However, it could benefit from more detailed explanations of the methodology, a deeper analysis of findings, and a more comprehensive presentation to provide a clearer picture of the research. Some of the issues which needs to be addressed before further process are the following:

Abstract:

1. This abstract need revision: Background and Gap (reason you are conducting this study in this area), are the main missing parts in this abstract.

Introduction:

2. The citation markers (e.g., "(1)", "(2-6)") disrupt the flow of the text and could be incorporated more smoothly. Please use a proper citation style.

3. The transition between discussing global food waste and focusing on Indonesia could be smoother. Consider using transitional phrases to connect these sections more effectively.

4. line 50, you said “Moreover, studies on FW in middle-income countries, particularly Southeast Asia, are limited.” However, you didn’t acknowledge the limitations of existing research. It's important to acknowledge the limitations of existing research, particularly the lack of detailed data on food waste in Indonesia. This could help set the stage for the current study and emphasize its importance.

You didn’t your study area and made all data underlying the findings in your manuscript fully available. Please do it.

Methodology:

5. While the study mentions the use of multistage random sampling, it does not elaborate on the specific details of how households were selected within each stage.

6. This approach introduces inconsistencies, potentially affecting data reliability. The diary method has limitations like forgetfulness and inaccurate estimations. I suggest to improve measurement techniques, you implement “waste audits”, where trained personnel sort and weigh household waste samples over an extended period. You can also use “Smart sensor technology”. This Smart sensor integrated into trash bins to automatically quantify and categorize waste, providing real-time data collection without relying on self-reporting.

7. The study mentions obtaining written informed consent from participants and approval from the Human Research Ethics Committee. However, it does not provide details on how privacy and confidentiality were ensured, especially given the sensitive nature of waste-related data.

8. The adaptation of the method from the National Zero Waste Council of Canada introduces a potential bias if the waste generation patterns and waste management practices in Indonesia differ significantly from those in Canada. This raises questions about the validity and applicability of the methodology to the local context.

Result:

9. Why can’t you present your result by figures? The figures presented in the comparison study could provide valuable insights into the differences and similarities between the urban and rural contexts.

10. Line 109. While statistical significance (p<0.001) is mentioned, it would be beneficial to provide confidence intervals or other measures of uncertainty to qualify the findings and indicate the robustness of the observed differences between urban and rural areas.

11. Line 135. Reasons for throwing away food:

Better by comparison: “why do urban areas generate more FW per capita? Are there specific consumption patterns, food distribution systems, or waste management practices that contribute to this disparity?”

Discussion:

12. The discussion does not provide a clear context for the study or its significance. It merely states the findings without explaining why the study was conducted or what implications these findings might have. It lacks depth in analysis and fails to offer comprehensive solutions to address the complex issue of food waste.

13. please cite your results (tables, figures) in this part

Conclusion:

14. This conclusion oversimplifies the complexities of FW generation and overlooks the importance of context-specific factors in determining effective prevention and reduction strategies. A more nuanced analysis taking into account the unique characteristics of each setting would provide a stronger foundation for developing targeted interventions to address FW in both urban and rural areas.

Reviewer #2: Dear author/s,

the topic of the manuscript is interesting, however there a few aspects that should be improved:

1. The research questions are missing, please add them in order to emphasize the originality of the manuscript.

2. So far the research instruments to determinate the reasons for food waste not presented. Whom fulfill it?

3. Which is the novelty of the research? What gap in the existing literature this study fills?

4. The discussion part could be improved by comparing your results with other similar researches.

5. Which are the limitations of the study? What about future research direction and managerial implications of the study?

Reviewer #3: I appreciate the work done by the authors. The manuscript provides a wealth of information and data demonstrating the FW in the study area. Through this literature, readers can learn about FW in Indian context, providing useful references and insights for practice and research in the field of FW management. For the above reasons, this manuscript is agreed for publication. However, there are some comments the authors have to address before the publication (See highlighted comments in the paper).

6. PLOS authors have the option to publish the peer review history of their article (what does this mean?). If published, this will include your full peer review and any attached files.

Reviewer #1: **Yes: **Dr. Vincent NZABARINDA

Reviewer #2: No

Reviewer #3: **Yes: **Dr. Emmanuel Bizimana

---

## [Author Response · Author response to Decision Letter 0]

9 May 2024

Respons to reviewers

Reviewer #1

This study focuses on analyzing the differences in the amount, types, and causes of household food waste between urban and rural areas. It was conducted in Bogor Regency, Indonesia, and involved 215 households. The researchers used waste compositional analysis for food and diaries for beverages to assess household food waste. However, it could benefit from more detailed explanations of the methodology, a deeper analysis of findings, and a more comprehensive presentation to provide a clearer picture of the research. Some of the issues which needs to be addressed before further process are the following:

Abstract:

1. This abstract need revision: Background and Gap (reason you are conducting this study in this area), are the main missing parts in this abstract.

Response: Thank you, we have added the background and gap in the abstract

Line 16-18: 

Studies on food waste in Southeast Asia are currently limited, with a notable absence of comparative analyses investigating the volume and composition of food waste in urban and rural areas through direct measurement.

Introduction:

2. The citation markers (e.g., "(1)", "(2-6)") disrupt the flow of the text and could be incorporated more smoothly. Please use a proper citation style.

Response: Thank you, we have changed it.

Line 35-38

Food loss and waste refer to a reduction in both the quantity and quality of food intended for human consumption along the food supply chain (1). Food waste (FW) occurs in retail (e.g., supermarkets and traditional markets), food services (e.g., restaurants, catering, canteens, and home consumption), households, and individuals (2–6).

3. The transition between discussing global food waste and focusing on Indonesia could be smoother. Consider using transitional phrases to connect these sections more effectively.

Response: Thank you, we have added the transition words. 

Line 39-52

The United Nations Environment Programme (UNEP) estimated that 17% of the food produced globally was discarded or not consumed. In 2019, the global average FW reached 121 kg/capita/year, equivalent to 931 million tons of food discarded at retail and consumer levels. Furthermore, households accounted for 61% of the total FW, while food services and retail contributed 26% and 13%, respectively (7).

Given Indonesia’s status as the fourth-most populous country in the world, understanding its food loss and waste trends become crucial. A study on food waste study in Indonesia over the two decades (2000-2019) revealed a concerning increase from 39% to 55%, equivalent to 5-19 million tons per year. Notably, 80% of this waste originated from households, with 44% being edible food (8). However, there remains a dearth of detailed data on the characteristics of FW in Indonesia, particularly through direct measurement. Halving global FW per capita at the retail and consumer levels is one of the targets of the Sustainable Developments Goals 12. To achieve this target, accurate FW measurements are essential for identifying the contributing factors.

4. line 50, you said “Moreover, studies on FW in middle-income countries, particularly Southeast Asia, are limited.” However, you didn’t acknowledge the limitations of existing research. It's important to acknowledge the limitations of existing research, particularly the lack of detailed data on food waste in Indonesia. This could help set the stage for the current study and emphasize its importance. You didn’t your study area and made all data underlying the findings in your manuscript fully available. Please do it.

Response: We have added the information.

Line 53-63

Studies on household FW have been conducted in many countries, especially developed countries in Europe (9–14), America (15,16), Canada (17,18), and China (19). The methods used to quantify food waste vary, including direct measurements such as waste composition analysis (11–13,18,20) and waste audits (17), as well as indirect measurements like modeling (16,21) and questionnaires (9,22). These studies also examined nutrient loss (15), factors related to food waste such as family characteristics (13,22), dietary quality (16,18), and consumer behaviors (9). However, a disproportionate number of studies have been conducted in urban areas than rural areas (7). Additionally, research on household FW quantity and composition in middle-income countries, particularly Southeast Asia, is limited. While, some studies utilize direct measurements of household FW, there is a lack of data on FW categorized by food types and groups, or comparisons between urban and rural areas (23–26).

Methodology:

5. While the study mentions the use of multistage random sampling, it does not elaborate on the specific details of how households were selected within each stage.

Response: Thank you for the comments, we have added the information in the method section

Line 96-100:

Multistage random sampling was employed to select the households. In the first stage, sub-districts with urban and rural characteristics were selected. In the second stage, villages or urban wards with the largest population were selected, and then neighborhoods within each village were randomly chosen. Finally, household samples were randomly selected from the list of households obtained from these selected neighborhoods.

6. This approach introduces inconsistencies, potentially affecting data reliability. The diary method has limitations like forgetfulness and inaccurate estimations. I suggest to improve measurement techniques, you implement “waste audits”, where trained personnel sort and weigh household waste samples over an extended period. You can also use “Smart sensor technology”. This Smart sensor integrated into trash bins to automatically quantify and categorize waste, providing real-time data collection without relying on self-reporting.

Response: Thank you for the suggestion. We might consider conducting waste audits in a separate study. Meanwhile, in the study conducted in 2023, we employed the diary method solely for beverage waste, while all solid waste was analyzed using waste compositional analysis. We discovered that beverage waste comprised only 0.8% of total food waste. Therefore, we are confident that this would not compromise the overall data reliability of the study.

7. The study mentions obtaining written informed consent from participants and approval from the Human Research Ethics Committee. However, it does not provide details on how privacy and confidentiality were ensured, especially given the sensitive nature of waste-related data.

Response: Thank you, we have already added confidentiality information regarding the research data.

Line 127-128

The data obtained from this research will be treated confidentially, only the research team responsible for this study will be able to access and use it.

8. The adaptation of the method from the National Zero Waste Council of Canada introduces a potential bias if the waste generation patterns and waste management practices in Indonesia differ significantly from those in Canada. This raises questions about the validity and applicability of the methodology to the local context.

Response: Thank you for pointing this out. We acknowledge that there are differences in waste generation patterns and waste management practices between Canada and Indonesia. Therefore, we adapted the extrapolation method by considering rural and urban characteristics specific to Indonesia, a step that was not taken in Canada.

The method adapted from the National Zero Waste Council of Canada involved extrapolation to calculate the average food waste at the regional level. In the method outlined by the National Zero Waste Council of Canada, estimating the quantity of food waste for a region could be conducted in several ways depending on the type of study carried out. The waste compositional study with individual sampling; extrapolation was calculated by multiplying the kilograms of food waste per household by the number of households in the area. In this study, the adaptation carried out was that the quantity of food waste had already been calculated in kilograms per capita per household in urban and rural areas, and extrapolation took into account the urban and rural populations proportionally. Therefore, the calculation for the average of food waste at the regional level was conducted by multiplying the kilograms of food waste per household in urban areas by the urban population, adding it to the kilograms of food waste per household in rural areas multiplied by the rural population, and then dividing by the total population.

Line 114-122:

In this study, the adaptation involved calculating the quantity of food waste in kilograms per capita per household in urban and rural areas, with extrapolation adjusting for the urban and rural populations proportionally. Therefore, the calculation for the average FW at the regional level was conducted by multiplying the kilograms of FW per household in urban areas by the urban population, adding it to the kilograms of FW per household in rural areas multiplied by the rural population, and then dividing by the total population. The total population of Bogor Regency at the end of 2022 was obtained from the Bogor Regency Population and Civil Registry Office. The total population in Bogor Regency was 5,427,068 people, consisting of 390,047 people in rural areas and 5,037,021 people in urban areas.

Result:

9. Why can’t you present your result by figures? The figures presented in the comparison study could provide valuable insights into the differences and similarities between the urban and rural contexts.

Response: Thank you, we have changed the table 1 into figures. The table were available in supplementary file. 

10. Line 109. While statistical significance (p<0.001) is mentioned, it would be beneficial to provide confidence intervals or other measures of uncertainty to qualify the findings and indicate the robustness of the observed differences between urban and rural areas.

Response: thank you, we have added the confidence intervals in the supplementary file.

11. Line 135. Reasons for throwing away food:

Better by comparison: “why do urban areas generate more FW per capita? Are there specific consumption patterns, food distribution systems, or waste management practices that contribute to this disparity?”

Response: Thank you for pointing this out. We have added it into the discussion section. Therefore, in the results section we only describe the results.

Discussion:

12. The discussion does not provide a clear context for the study or its significance. It merely states the findings without explaining why the study was conducted or what implications these findings might have. It lacks depth in analysis and fails to offer comprehensive solutions to address the complex issue of food waste.

Response: Thank you for pointing this out. We have revised the discussion section. Implication of the findings also provided in the policy implication.

The most commonly discarded edible FW in both urban and rural areas consist of cereals, tubers and their derivatives, vegetables, and legumes (Figure 2). The main contributors to inedible food waste were fruits and their derivatives, along with vegetables (Figure 3). The results of this study imply that strategies for preventing and reducing household FW can focus on four food groups. In urban areas, education can be concentrated on preventing and reducing edible FW (rice, vegetables, tempeh, and tofu) as well as practical training to reuse or recycle inedible FW (fruits and vegetables). Meanwhile, in rural areas, education can be focus reducing edible FW (vegetables) by improving storing practices and practical training to reuse or recycle inedible FW (fruits).

Various reasons underlie food waste, depending on the type of food (Table 3). Information on food storage, including techniques to extend the shelf life of food and proper food storage practices, is applicable to all types of food. Encouraging behavioral changes in cooking and eating habits is crucial to preventing and reducing food waste, especially for foods such as rice, vegetables, plant-based foods (tofu and tempeh), and cooked foods. Utilizing data-related information on food packaging can help prevent wastage of bread and packaged foods.

The proportion of inedible FW was higher than that of edible FW in both the urban and rural areas. Urban households were more likely to discard edible FW than rural households (38.2% and 23.4 %, respectively) (Figure 1). This implies that urban households throw away more food suitable for consumption. Consistent with studies in the European Union and Lebanon, this study found that the quantity of household FW in urban areas was higher than that in rural areas (32,33). 

Household food waste generated in urban areas exceeds that of rural areas, attributed to disparities in food systems, food access, consumer behavior, and food waste utilization or management. In rural settings, self-produced food sources prevail, contrasting with the urban areas where processed food purchases predominate, leading to amplified food waste generation (34). The growth of modern retail, restaurants, and fast-food outlets in urban areas affects consumer preferences, purchasing behavior, and food accessibility, thereby driving higher food waste rates (35). Urban residents tend to prefer quick and convenient food preparation, resulting in over-purchasing, high consumption of processed foods, and foods from outside the home (restaurants, fast food outlets, and street food) (34). This trend contributes to the generation of food waste (34,36). On the other hand, many rural households convert their food waste into compost or use it as feed for pets or livestock (38,39), thereby reducing the amount of FW disposed of in bins (38).

This study found a higher household income and education level in urban areas than in rural areas (Table 1). Income is a major driver to household FW generation (22). Urban residents have higher incomes but allocate a lower proportion of food expenditure than those living in rural areas, leading to a higher dietary diversity (35). Moreover, the higher dietary quality correlates with greater FW, as individuals with a high dietary quality consume larger quantities of vegetables and fruits, which are the most discarded food waste (37). 

13. please cite your results (tables, figures) in this part

Response: Thank you, we have cited tie tables or figures in the discussion section

Conclusion:

14. This conclusion oversimplifies the complexities of FW generation and overlooks the importance of context-specific factors in determining effective prevention and reduction strategies. A more nuanced analysis taking into account the unique characteristics of each setting would provide a stronger foundation for developing targeted interventions to address FW in both urban and rural areas.

Response: Thank you for the comments. We have revised the conclusion.

Line 273-287:

The quantity of household food waste differs significantly between urban and rural areas, with urban households discarding more edible food than their rural counterparts. Cereals, tubers and their derivatives (especially rice), vegetables, and legumes were the major contributors to edible food waste, whereas fruits were the main contributors to inedible food waste in both areas. Additionally, the drivers of food waste were found to be similar in both urban and rural areas.

Strategies for food waste prevention and reduction should target urban areas and focus on food groups that contribute significantly to edible food waste, such as cereals, tubers, and their derivatives, particularly rice, vegetables, and legumes. Meanwhile, managing inedible food waste should prioritize the reuse and recycling of fruits and vegetables to prevent their disposal in landfills, especially in rural areas.

Disparities exist in food waste drivers, primarily depending on the type of food rather than the location of the resident. Therefore, awareness campaigns and behavior change initiatives should take into account the main drivers of food waste based on the type of food. Understanding the quantity, composition, and driver

---

## [Editor Report · Decision Letter 1]

24 May 2024

The quantity and composition of household food waste: implications for policy

PONE-D-24-03754R1

Dear Dr. Diana,

We’re pleased to inform you that your manuscript has been judged scientifically suitable for publication and will be formally accepted for publication once it meets all outstanding technical requirements.

Kind regards,

Fabien MUHIRWA

Academic Editor

PLOS ONE

---

## [Editor Report · Acceptance letter]

3 Jun 2024

PONE-D-24-03754R1 

PLOS ONE

Dear Dr. Diana, 

I'm pleased to inform you that your manuscript has been deemed suitable for publication in PLOS ONE. Congratulations! Your manuscript is now being handed over to our production team.

Kind regards, 

on behalf of

Dr. Fabien MUHIRWA 

Academic Editor

PLOS ONE